# Effect of the Iron Reduction Index on the Mechanical and Chemical Properties of Continuous Basalt Fiber

**DOI:** 10.3390/ma12152472

**Published:** 2019-08-03

**Authors:** Lida Luo, Qichang Zhang, Qingwei Wang, Jiwen Xiao, Jin Liu, Linfeng Ding, Weizhong Jiang

**Affiliations:** 1State Key Laboratory for Modification of Chemical Fibers and Polymer Materials, Donghua University, Shanghai 201620, China; 2Engineering Research Center of Advanced Glass Manufacturing Technology, Ministry of Education, Donghua University, Shanghai 201620, China

**Keywords:** Basalt fiber, iron redox index, spectral photometric method, tensile strength, chemical durability.

## Abstract

Basalt glass belongs to the iron-rich aluminosilicate glass system; thus, the iron content and the iron redox index (IRI=Fe^2+^/Fe_total_) influence the viscosity, density, mechanical and chemical properties of basalt fiber (BF). In this work, continuous BFs with IRIs ranging from 0.21–0.87 were prepared by adding a different amount of redox agents. An economical and easily accessible testing method—the spectral photometric method with 1,10-phenanthroline—is applied to measure the IRI with convinced accuracy, which has been approved by Mössbauer spectra and X-ray fluorescence analysis. The tensile strength of the BF samples increases approximately linearly with increasing IRI as a function of σ=227.9IRI+780.0. The FT-IR results indicate that, with increasing IRI, the ferric ions are replaced by the much stronger network formers (Al^3+^ and Si^4+^), hence the increased the tensile strength. The X-ray diffraction results show an amorphous nature of BF samples. Moreover, the tensile strength is significantly decreased after the alkali corrosion, which is partly attributed to the severe surface damaging according to the SEM results. This work proved the feasibility of mechanical property improvement in BF production by controlling the iron redox index.

## 1. Introduction

Basalt fiber (BF) is an excellent reinforcing material and has been widely applied in the manufacturing industry due to its advantages, like good mechanical properties, high chemical and thermal stability, and natural abundance, etc. [1,2,3]. Recent research [3] has demonstrated that the high tensile strength of BF results from the high Fe_2_O_3_ content; thus, the role of iron in basalt glass and fiber has attracted attention and has been studied widely.

Iron is present in the ferrous and ferric states in BF [4]. The iron reduction index (IRI, IRI = Fe^2+^/Fe_total_) refers to the proportion of ferrous to total iron in the material. It is generally believed that Fe^3+^ ions are tetrahedrally coordinated network former cations while Fe^2+^ ions with octahedral coordination act as network modifier [5,6]. 

The redox state (coordination and content) of iron in BF determines both the fiber properties (tensile strength, thermal stability, and crystallization behavior [7,8,9]), the fabrication parameters (temperature and viscosity [10]), light absorption [11], and magnetic properties [12]. For example, Gutnikov et al. [3] studied the effect of the reduction treatment on the BF crystallization properties and found out the glass transition temperature decreases after the reduction and the crystallization starts at a lower temperature in reduced BF. Jensen et al. [13] applied the X-ray photoelectron spectroscopy (XPS) to determine the IRI in a multicomponent glass and demonstrated that the IRI decreases with increasing melt temperature due to an entropy-driven reduction. Yue et al. [14] demonstrated that the control of the redox state of stone wool fibers is crucial for the fire barrier function.

The oxidation process has been studied widely in geoscience [15], as well as in glass science [14]. The oxygen gradient between the glass and the environmental atmosphere dominates the oxidation process. Smedskjaer et al. [16] heat-treated three types of amorphous stone wool fibers under oxidizing conditions and demonstrated that the outstanding high-temperature stability of samples is attributed to the oxidation of Fe^2+^ to Fe^3+^. The oxidation process in glass fibers will form a nanocrystalline surface layer and increase the viscosity [16,17]. 

A fundamental understanding of the effect of the iron redox ratio on the mechanical and chemical properties of basalt fiber is critically needed to guide new product development and associated economics of the basalt fiber industry [18], including the control of batch material and melting atmosphere. In this work, continuous basalt fibers were successfully prepared. The ratio of ferric oxide and ferrous oxide in the BF was changed by adding redox agents [19] to the melting atmosphere. The effect of the IRI on the mechanical and chemical properties of basalt fibers was studied.

## 2. Experimental Procedures

### 2.1. Materials

The basalt fibers (BFs) were prepared by using basalt rocks from Shandong Province (China) and additional redox agents. The adding of MnO_2_ (oxidation), carbon powder (reduction), and citric acid (reduction) will change the environmental atmosphere in the furnace and thus change the IRI of basalt rocks. The chemical composition of the basalt rocks (determined by XRF) are shown in Table 1. 

### 2.2. Sample Preparation

Basalt rocks were cracked into particles by a planetary ball mill (QM-1SP2-CL, China) and sieved by using an 80-mesh sieve to narrow the particle size distribution. Then, different amounts of redox agents (MnO_2_, carbon powder and citric acid) were added into the basalt powder. Afterward, the basalt mixture was placed in a quartz crucible, heated to 1550 °C at a heating rate of ~600 °C/h in a high-temperature furnace, with an isothermal hold for 2 h. The molten basalt mixture was rapidly quenched in water to get the redox-treated basalt glass. 

The continuous BFs were prepared by using a laboratory monofilament fiber drawing system [3,9] with a platinum crucible at a temperature ~1350 ± 10 °C and a take-up reel at a rotation speed of ~240 rpm.

### 2.3. Characterization Techniques

The total iron content and ferrous content were measured by applying a spectral absorption spectrometer (722N, Shanghai, China). We followed the ISO 14719-2011 standards by mixing the BF powers in 1,10-phenanthroline solution to investigate the ferrous and total iron content. To double-check the measurement accuracy, we applied the Transmission ^57^Fe Mössbauer spectra (MFD-500AV-01, Tokyo, Japan) to check the ferrous content and applied X-ray fluorescence analysis (Axios, Almelo, The Netherlands) to check the total iron content.

The tensile strength of the BFs was measured by using the Single Fiber Electronic Tensile Strength Tester (XQ-1A, Wenzhou, China). The chemical structure was measured by applying Fourier transform infrared (FT-IR, Nexus-670, Waltham, MA, USA) spectra, where the BFs samples were mounted in the KBr pelletized disks and the surface of samples were measured in the 400–1400 cm^−1^ region with a resolution of 2 cm^−1^. The micromorphology of the samples was studied by using field emission scanning electron microscope (SEM, SU8010, Tokyo, Japan). The crystalline phases analysis was conducted by means of X-ray diffraction (D/max-2550PC, Tokyo, Japan).

The chemical durability was assessed by leaching tests [18] conducted in 1–3 mol/L HCl solutions and 0.5–1.5 mol/L NaOH solutions at 90 °C for 12 h. 

## 3. Results 

### 3.1. Iron Redox State

The iron reduction indexes (IRI) reported in Table 1 were investigated and calculated by the spectral photometric method. The results show that the adding of redox agents had effectively changed the IRI. The IRI of basalt fiber (BF) without any redox agent was 0.31, which was higher than the BF prepared by adding 3 wt % MnO_2_ (IRI = 0.21) and much lower than the BF prepared by adding 1.5 wt % carbon (IRI = 0.87). 

Mössbauer spectra are widely used to determine the ferrous and ferric content in glass science [3] with an accuracy of around 10%, while the cost is much higher than the spectral photometric method. Thus, we applied the Mössbauer spectra to double-check the measurement accuracy of the spectral photometric method. It can be seen in Figure 1 that the IRI5 BF (adding 1.0 wt % citric acid) had a Fe^2+^/Fe^3+^ ratio of 64/36 (IRI = 0.64), which was equal to the result from the spectral photometric method. Moreover, we also applied the X-ray fluorescence analysis to check the total iron content of the BF and the uncertainty was within 2%. 

### 3.2. Tensile Strength and FT-IR

The tensile strength of the BF samples in this work was measured under the same conditions at room temperature and the average values of each of the 20 samples are reported in Figure 2a. The measurement uncertainty was calculated by using a *t*-test probability with a 95% confidence interval. The tensile strength increased approximately linearly with increasing IRI, which can be fitted by the excel linear function:(1)σ=227.9IRI+780.0
where σ is the average tensile strength in MPa and IRI is the iron reduction index. The digital photos, as shown in Figure 2a, also indicated that the color of the BFs (from the visual investigation) became darker with increasing IRI.

It can be found in Figure 2b that the infrared absorption peaks of basalt fibers with different iron reduction index were distinct from each other, especially, the broadband around 750 cm^−1^, which was enhanced with increasing IRI and a sharp peak was observed around the 1035 cm^−1^ region.

### 3.3. XRD

All BF samples were investigated by XRD and the results shown in Figure 3 indicated that all the fibers were amorphous. Hence, the tensile strength of BF prepared in this work will not be influenced by crystallization during the processing.

### 3.4. Chemical Durability

As can be seen in Figure 4, the tensile strength significantly decreased after chemical corrosion. Notably, the IRI7 sample only had 34% tensile strength retention after 1.5 mol/L NaOH corrosion and tensile strength 70% retention after 3 mol/L HCl corrosion. The tensile strength and mass retention also decreased with the increase of HCl (1–3 mol/L) and NaOH (0.5–1.5 mol/L) proportions, where the overall shape of the curves kept the same, respectively. Particularly, both the tensile strength and mass retention decreased with increasing IRI in alkali solutions, while not obvious in acid solutions. 

According to the SEM image in Figure 5a, the BF had an average diameter of around 37 ± 2 μm and the surface was quite smooth, without any presence of defects from the visual investigation. However, in Figure 5b, BF sample after 0.5 mol/L NaOH corrosion showed severe surface damaging with a large amount of big spherical particles and a large area of surface peeling appeared on the surface. 

## 4. Discussion

Regarding the iron redox state, the spectral photometric method used in this work was confirmed by the Mössbauer spectra and X-ray fluorescence analysis with an uncertainty less than 2%, thus, the IRI measured by the spectrophotometric method has accuracy. The spectral photometric method is a much cheaper and easier accessible method compared to the Mössbauer spectra and X-ray photoelectron spectroscopy. 

For the relationship between tensile strength and structure, the tensile strength was strongly associated with the network structure, which could be proved by the FT-IR results. The broadband around 750 cm^−1^, which enhanced with increasing IRI, was attributable to the symmetric stretching vibrations of the Si(Al)–O bond in the [Si(Al)O_4_] tetrahedral structure [20]. And the sharp peak around the 1035 cm^−1^ region was connected with the Si–O–Si and Si–O–Al bridging oxygen vibration [21]. The ferric ions (Fe^3+^, act as a network former) were replaced by the much stronger network formers (Al^3+^ and Si^4+^). Therefore, the enhancement of the IR absorption around 756 cm^-1^ indicated a much stronger glass network, which was the reason for the increase of tensile strength with increasing IRI. However, further studies are required to quantify the changing of the tetrahedral structure. 

Concerning the chemical durability, previous research [19,22] showed that a silicate gel was found on the fiber surfaces, which acted as a protective layer and slowed down the kinetics of the acid attack. However, in the alkali solution, the fiber surface contained fractures and severe damaging without any protection from a possible passivation layer. The defects on the BF surface observed in Figure 5b could be one of the reasons for the big reduced tensile strength after alkali corrosion. The strong hydrolysis of the Si–O–Si bond, Si–O–Al bonds, and/or Al–O–Al bonds by OH^-^ groups in the alkali solution [19] was attributed to the decreasing of tensile strength retention with increasing IRI.

## 5. Conclusion

Continuous basalt fibers (BF) with iron reduction indexes (IRI) from 0.21–0.87 were prepared and the effect of the IRI on the mechanical and chemical properties was studied in this work. Firstly, the spectral photometric method with 1,10-phenanthroline was tested to measure the IRI with accuracy, which was approved by Mössbauer spectra and XRF. The spectral photometric method is a more economical and easier accessible method, recommended to study the IRI of BF. Secondly, the tensile strength of BF samples increased approximately linearly with increasing IRI as a function of σ=227.9IRI+780.0. The FT-IR indicated that the ferric ions were replaced by the much stronger network formers (Al^3+^ and Si^4+^) and increased the tensile strength. The idea of controlling the iron redox index during processing will be a good approach for the mechanical property improvement of BF production. Lastly, the tensile strength was significantly decreased after the alkali corrosion, which was more than double after acid corrosion. Particularly, the tensile strength retention decreased with increasing IRI. The SEM results indicated that the great tensile strength decrease after alkali corrosion was partly attributed to the severe surface damaging. We note that the improvement of tensile strength by increasing the IRI results in a slight reduction of the anti-alkali corrosion.

## Figures and Tables

**Figure 1 materials-12-02472-f001:**
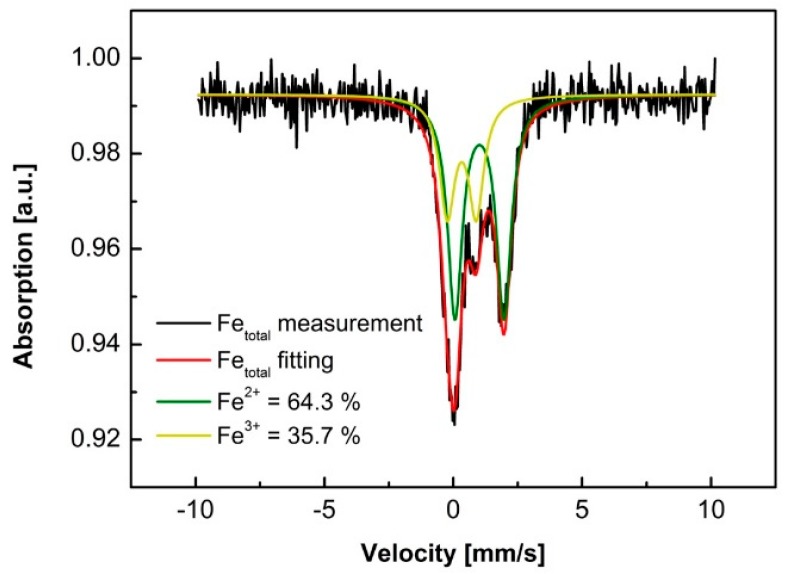
^57^Fe resonant absorption Mössbauer spectra of IRI5 basalt fiber sample (iron redox index = 0.64).

**Figure 2 materials-12-02472-f002:**
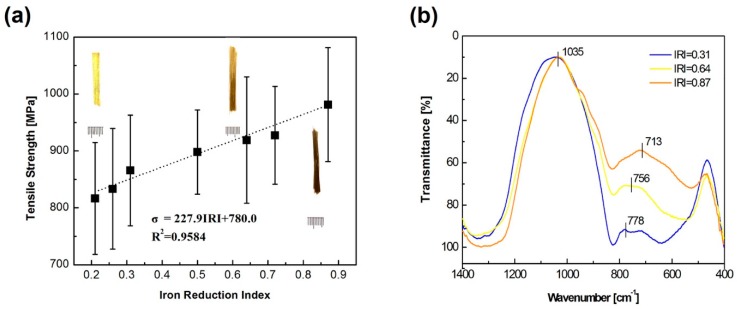
(**a**) Tensile strength of basalt fiber samples; (**b**) FT-IR absorption spectra of basalt fiber samples.

**Figure 3 materials-12-02472-f003:**
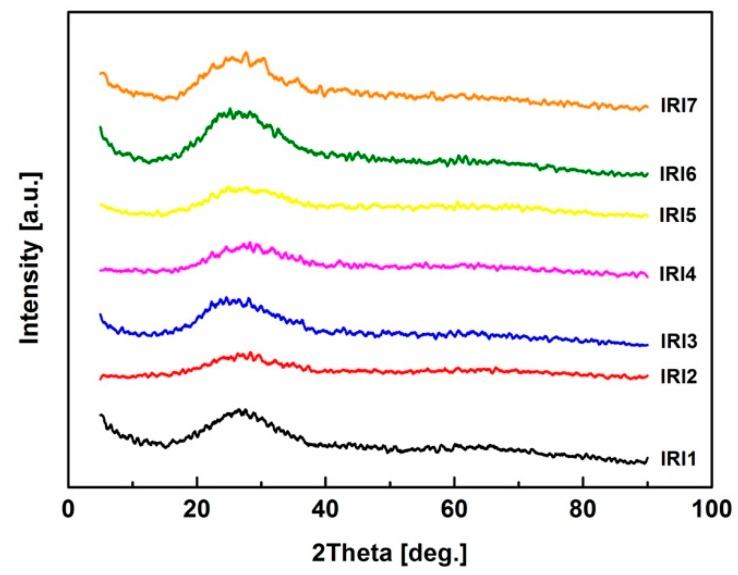
XRD patterns of basalt fiber samples.

**Figure 4 materials-12-02472-f004:**
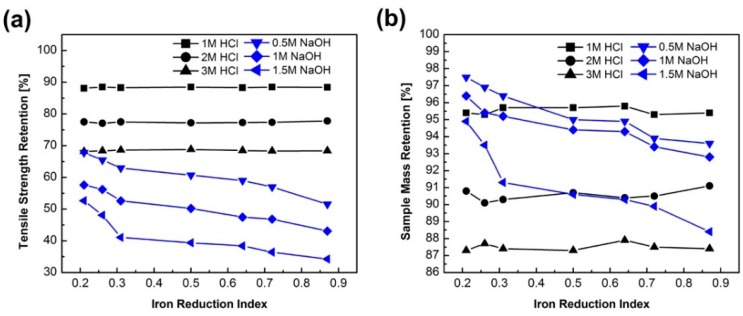
(**a**) Tensile strength retention of basalt fiber samples after alkali corrosion; (**b**) mass retention of basalt fiber samples after acid corrosion.

**Figure 5 materials-12-02472-f005:**
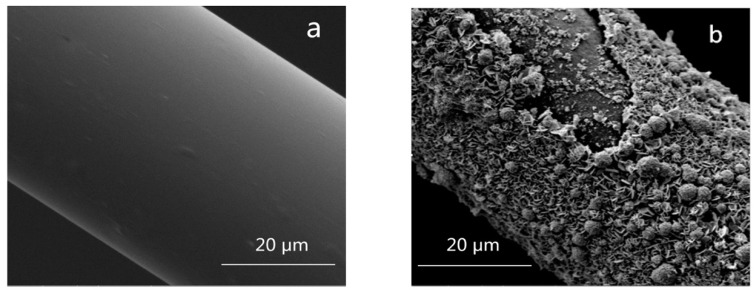
(**a**) SEM image of basalt fiber IRI3 sample; (**b**) SEM image of basalt fiber IRI3 sample after 0.5 mol/L NaOH corrosion.

**Table 1 materials-12-02472-t001:** Chemical compositions of basalt mixture (wt %).

Samples	SiO_2_	Al_2_O_3_	Fe_2_O_3_	CaO+MgO	K_2_O+Na_2_O	Others	MnO_2_	Citric Acid	C	IRI
IRI1	55.17	15.57	9.10	12.23	5.89	2.04	3.00	/	/	0.21
IRI2	55.17	15.57	9.10	12.23	5.89	2.04	1.00	/	/	0.26
IRI3	55.17	15.57	9.10	12.23	5.89	2.04	/	/	/	0.31
IRI4	55.17	15.57	9.10	12.23	5.89	2.04	/	0.50	/	0.50
IRI5	55.17	15.57	9.10	12.23	5.89	2.04	/	1.00	/	0.64
IRI6	55.17	15.57	9.10	12.23	5.89	2.04	/	/	0.50	0.72
IRI7	55.17	15.57	9.10	12.23	5.89	2.04	/	/	1.50	0.87

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
