# Peer review of "Effect of the Iron Reduction Index on the Mechanical and Chemical Properties of Continuous Basalt Fiber"

_materials, 2019, doi:10.3390/ma12152472_

Round 1

Reviewer 1 Report

It’s an interesting research, however, it’s easy to find lots of drawbacks. The authors may very carefully revise the manuscript.

1) Introduction is too short.

2) Not only tensile strength, modulus is a very critical parameter, which should be included into the discussion.

3) 3.4. Chemical durability. This section should be extended. There’s no solid explanation for the reduction of tensile strength. Many tests can be employed here, for instance, SEM may monitor the surface changes after corrosion, XPS may detect the chemical property changes on the surface. Based on the analysis, the authors may provide a stuff (like a chemical reaction) to demonstrate those changes.

4) 4. Discussion. The authors explained the reason for loosing strengths after being corroded in NaOH, however, only cite others’ research is not enough. Please provide solid evidences yourselves.

5) In Conclusion, the authors claimed that XRD and SEM results show an amorphous nature of BF samples. I don’t believe that people can see amorphous from SEM image (Figure 3b).

6) Language can be improved.

Author Response

Dear Reviewer,

Thank you for your review of our paper and the comments on our manuscript. The suggestions you gave in the comments are very helpful to develop the manuscript. We have answered each of your comments below in blue and revised the manuscript accordingly in red.

1)   Introduction is too short.

Response: We revised the manuscript in Lines 34-53. We did some more literature screen and add 10 more relevant references into the manuscript. We especially highlighted more on the research on the oxidation process which will support our experimental method.

2)   Not only tensile strength, modulus is a very critical parameter, which should be included into the discussion.

Response: Sure, elastic modulus is a very critical parameter for basalt fiber. While we don’t have elastic modulus data in our manuscript, thus, we feel not confident to discuss the modulus in our manuscript.

3) 3.4. Chemical durability. This section should be extended. There’s no solid explanation for the reduction of tensile strength. Many tests can be employed here, for instance, SEM may monitor the surface changes after corrosion, XPS may detect the chemical property changes on the surface. Based on the analysis, the authors may provide a stuff (like a chemical reaction) to demonstrate those changes.

Response: We did some extra SEM measurements and revised the manuscript in Lines 131-140. A new Figure 5 is added to the manuscript.

4)  4. Discussion. The authors explained the reason for loosing strengths after being corroded in NaOH, however, only cite others’ research is not enough. Please provide solid evidences yourselves.

Response: We revised the manuscript in Lines 155-167. A new Figure 5 is added to the manuscript.

5) In Conclusion, the authors claimed that XRD and SEM results show an amorphous nature of BF samples. I don’t believe that people can see amorphous from SEM image (Figure 3b).

Response: The reviewer is correct. We deleted the misleading sentence.

6) Language can be improved.

Response: Some minor changes are made in the revised manuscript.

We hope the modifications have improved our manuscript!

Reviewer 2 Report

Dear Authors, considering the amount of information and volume of your paper it is more like Letter and not an Article. In case you plan to publish it as Article then you need to add more information into your Introduction, add some more results, improve the quality of your charts and results description, add more info in your discussions and conclusions; also to make obvious novelty of your investigation through the text in comparison to existing techniques and achieved results by other researchers; increase amount of your references.

Author Response

Dear Reviewer,

 Thank you for your review of our paper and the comments on our manuscript.

We have revised the manuscript accordingly in red.

We hope the modifications have improved our manuscript!

Reviewer 3 Report

Overall a good piece of research. The writing was for the most part good however just needs a little more care with grammar. Very good overall. I would take care when using words such as ‘might’ and ‘may’ when talking about results as it gives the impression of uncertainty. I would request that language be more definitive or if not possible, state your Hypothosis and outline that further research is needed. The specific question of why the tensile strength increases should be discussed in more depth (line 127/129).

Minor Comments:

Line 12: ‘Basalt glass belong to’

Line 18: ‘Might’. This term seems non definitive. What is the cause of ferric ion replacement? If it is the replacement of aluminium ions then ‘might’ should be removed.

Line 27: ‘The industry’: what industry? Please make clear.

Line 29: One reference I believe is insufficient for such a large description of the beneficial properties of BFs. Need more references where [1] is currently.

Line 53: Basalt rocks were cracked

Line 53: How were the rocks cracked into particles? Please be specific and list the tools used to crack these said rocks.

Line 57: Restructure sentence: ‘The molten basalt mixture was rapidly quenched in water to get the redox-treated basalt glass.’

Line 60/61: What is the diameter tolerance of the fibers? If known, please add? How accurate can the fiber diameter be controlled?

Line 65: ‘… to investigate the ferrous and total iron content’

Line 70: FT-IR was used to measure microstructure. Does it not measure the chemical structure? Please reword. Also, does this sentence mean the cross section of the BF or simply the surface? Please be more specific. Similarly, how was the fiber prepared/mounted for FT-IR and what tips were used (e.g. diamond, germanium)? Please specify.

Line 71: SEM was used to measure micromorphology. Was the SEM able to give you a roughness value or was this purely a visual investigation? Please specify.

Line 84: ‘Trustable’ isn’t the best word. Maybe remove and just leave accuracy.

Line 94: “Student’s t-test” is not appropriate. Reword: The measurement uncertainty was calculated by using a t-test probability with a 95% confidence interval.

Line 97: Please add a coefficient of determination to the linear equation associated with figure 2.a.

Line 100: ‘…cm-1 which is enhanced with increasing IRI’

Line 123: ‘Double checked’ isn’t definitive language. Change to ‘confirmed by’.

Line 128: ‘which may’? Is it or is it not attributed to the symmetric stretching vibrations of the Al-O bond in the tetrahedral structure? Need a specific answer.

Line 148: ‘Increases approximately linearly’ at a rate of… (please specify the rate of linear increase).

Author Response

Dear Reviewer,

First of all, please don’t surprise on our fast handling, we already got the reviews of another two anonymous reviewers weeks ago. Thank you for your review of our paper and the comments on our manuscript. We greatly appreciate your positive general comments. The suggestions you gave in the comments are very helpful to develop the manuscript. We have answered each of your comments below in blue and revised the manuscript accordingly in red.

Comments:

Overall a good piece of research. The writing was for the most part good however just needs a little more care with grammar. Very good overall. I would take care when using words such as ‘might’ and ‘may’ when talking about results as it gives the impression of uncertainty. I would request that language be more definitive or if not possible, state your Hypothosis and outline that further research is needed. The specific question of why the tensile strength increases should be discussed in more depth (line 127/129).

Response: Obviously the reviewer already caught the most important point in our manuscript. We took the advice to explain what is our hypothesis and outlined the further researches needed. We revised the manuscript in Line 155-162.

Line 12: ‘Basalt glass belong to’

Response: We revised the manuscript in Line 12.

Line 18: ‘Might’. This term seems non definitive. What is the cause of ferric ion replacement? If it is the replacement of aluminium ions then ‘might’ should be removed.

Response: We revised the manuscript in Line 18-20.

Line 27: ‘The industry’: what industry? Please make clear.

Response: We added a word “manufacturing”.

Line 29: One reference I believe is insufficient for such a large description of the beneficial properties of BFs. Need more references where [1] is currently.

Response: We added 2 more references.

Line 53: Basalt rocks were cracked; How were the rocks cracked into particles? Please be specific and list the tools used to crack these said rocks.

Response: We revised the manuscript in Line 68. Basalt rocks were cracked into particles by a planetary ball mill (QM-1SP2-CL, China).

Line 57: Restructure sentence: ‘The molten basalt mixture was rapidly quenched in water to get the redox-treated basalt glass.’

Response: We revised the manuscript in Line 72-73. Thanks for your correction.

Line 60/61: What is the diameter tolerance of the fibers? If known, please add? How accurate can the fiber diameter be controlled?

Response: We revised in the manuscript in 75-76. The diameter of basalt fibers prepared in our work with a uncertainty around 2 μm which we described in Line 131 for Fig.5 (a).

Line 65: ‘… to investigate the ferrous and total iron content’

Response: We revised in the manuscript in Line 80.

Line 70: FT-IR was used to measure microstructure. Does it not measure the chemical structure? Please reword. Also, does this sentence mean the cross section of the BF or simply the surface? Please be more specific. Similarly, how was the fiber prepared/mounted for FT-IR and what tips were used (e.g. diamond, germanium)? Please specify.

Response: We revised the manuscript in 85-87.

Line 71: SEM was used to measure micromorphology. Was the SEM able to give you a roughness value or was this purely a visual investigation? Please specify.

Response: We revised the manuscript in Line 138-141. We specified it as “…from the visual investigation”. We also add another SEM imagine in Fig.5 (b) to support our hypothesis.

Line 84: ‘Trustable’ isn’t the best word. Maybe remove and just leave accuracy.

Response: We removed this misleading word.

Line 94: “Student’s t-test” is not appropriate. Reword: The measurement uncertainty was calculated by using a t-test probability with a 95% confidence interval.

Response: We revised the manuscript in Line 112.

Line 97: Please add a coefficient of determination to the linear equation associated with figure 2.a.

Response: We revised the Fig.2 (a).

Line 100: ‘…cm-1 which is enhanced with increasing IRI’

Response: We revised the manuscript in Line 119-120.

Line 123: ‘Double checked’ isn’t definitive language. Change to ‘confirmed by’.

Response: We revised the manuscript in Line 150.

Line 128: ‘which may’? Is it or is it not attributed to the symmetric stretching vibrations of the Al-O bond in the tetrahedral structure? Need a specific answer.

Response: We revised the manuscript in Line 155-162. We gave a much clearer statement in the revised manuscript.

Line 148: ‘Increases approximately linearly’ at a rate of… (please specify the rate of linear increase).

Response: We revised the manuscript in Line 177.

We hope the modifications have improved our manuscript!

Reviewer 4 Report

The paper present researches relate to the effect of the iron reduction index on the mechanical and chemical properties of continuous basalt fiber.

From the analysis of the information presented in the paper I found the following aspects:

- the paper, through content, presents a series of information that could be useful to the scientific community;

- the introduction part is very succinte and needs to be filled in with other newer information in the field;

- the choice for experimentation of the 7 samples presented in Table 1 should be justified;

- it must be presented images with the made samples

- for equation 1, the manner in which this has been achieved must be specified and, if taken from another source, a bibliographic source should be specified;

- for the chemical durability analysis HCl and NaOH were used in varying proportions. These proportions must be justified;

- a linear model is presented in Figure 2.a, but this is not the most representative;

- the conclusions section begins with a repetition of the abstract and should be more focused on the practical applicability of the results of the researches carried out;

- from the analysis of the data specified in the paper it follows that a series of information must also be added which show the novelty of the study;

- on the basis of the specifications, I consider that the paper can only be published after the changes have been made.

Author Response

Dear Reviewer,

First of all, please don’t surprise on our fast handling, we already got the reviews of another two anonymous reviewers weeks ago. Thank you for your review of our paper and the comments on our manuscript. We greatly appreciate your positive general comments. The suggestions you gave in the comments are very helpful to develop the manuscript. We have answered each of your comments below in blue and revised the manuscript accordingly in red.

Comments:

The paper present researches relate to the effect of the iron reduction index on the mechanical and chemical properties of continuous basalt fiber.

From the analysis of the information presented in the paper I found the following aspects:

- the paper, through content, presents a series of information that could be useful to the scientific community;

Response: We appreciate your positive comments.

- the introduction part is very succinte and needs to be filled in with other newer information in the field;

Response: We revised the manuscript by adding new references. Line 34-53.

- the choice for experimentation of the 7 samples presented in Table 1 should be justified;

Response: We revised the manuscript by adding clarification of the redox agents. Line 63-65.

- it must be presented images with the made samples

Response: We add three reference digital photos in Fig.2(a) and some descriptions into the revised manuscript. Line 115-117.

- for equation 1, the manner in which this has been achieved must be specified and, if taken from another source, a bibliographic source should be specified;

Response: We revised the manuscript and Fig.2(a). The fitting is directly from excel and we add the coefficient of determination to the figure. Line 113-114.

- for the chemical durability analysis HCl and NaOH were used in varying proportions. These proportions must be justified;

Response: We revised the manuscript with more descriptions. Line 133-135.

- a linear model is presented in Figure 2.a, but this is not the most representative;

Response: We revised the manuscript and Fig.2(a). The fitting is directly from excel and we add the coefficient of determination to the figure. Line 113-114.

- the conclusions section begins with a repetition of the abstract and should be more focused on the practical applicability of the results of the researches carried out;

Response: We revised the manuscript. Line 173-184.

- from the analysis of the data specified in the paper it follows that a series of information must also be added which show the novelty of the study;

Response: We revised the manuscript especially the abstract part. Line 15-24.

- on the basis of the specifications, I consider that the paper can only be published after the changes have been made.

We hope the modifications have improved our manuscript!

Round 2

Reviewer 1 Report

Accept.

Reviewer 2 Report

Dear Authors, your introduction can be improved.

Reviewer 4 Report

The authors responded to the observations made. The paper can be published in this form